# New Findings in the Field of Thermal Drilling of Aluminum Alloys

**DOI:** 10.3390/ma13215007

**Published:** 2020-11-06

**Authors:** Lydia Sobotova, Miroslav Badida, Marek Moravec, Anna Badidova, Alica Maslejova

**Affiliations:** 1Faculty of Mechanical Engineering, Technical University of Kosice, Letna 9, 040 01 Kosice, Slovakia; miroslav.badida@tuke.sk (M.B.); marek.moravec@tuke.sk (M.M.); anna.badidova@tuke.sk (A.B.); 2U. S. Steel Kosice, s.r.o., 044 54 Kosice, Slovakia; AMaslejova@sk.uss.com

**Keywords:** material joining, thermal drilling, visual evaluation, microstructure

## Abstract

This paper explores the joining of materials by a new progressive method of thermal drilling. Many types of joints are utilized in industrial production, especially in the automotive industry, which requires the joining of different types of materials with different thicknesses. For these needs, it will be appropriate to apply the joining method of thermal drilling technology. The used material, its mechanical properties, and the preparation of the joint by assembly and disassembly operations play an equally important role. By using new friction technologies, we can reduce production time, increase joint quality, offer automation in certain types of operations, save economic costs, and protect the environment. In this contribution, the authors present the results of their scientific research work focused on an investigation, comparison and testing of thermal drilling effects on the behavior of aluminum alloys (AlMgSi). The holes, collars and bushings formed by the drilling method were subjected to visual shape evaluation. Based on the evaluation of the samples, the quality of the evaluated joints was assessed. The best shape and strength properties at higher speeds of the drilling machine were obtained at 3400 rpm. The results of the methods of macroscopy and microscopy investigations, as well as the results of the methods of scanning electron microscope (SEM) and energy dispersive X-ray (EDS), are presented in this contribution.

## 1. Introduction

The automotive industry, together with other production industries, will rapidly include new progressive and useful manufacturing processes. With the development and use of new materials arise the requirement of special assembly of materials in the production of automobiles.

Therefore, in the theory of “design for assembly (DfA)” the author of [1] defined the process as a manufactured product that is designed to be quickly and easily assembled. Such design shape simplifications enable assembly with a reduced number of operations, also improving the manipulation of the components, and they can also change or modulate the required operations. During the assembly, techniques are used [1] that can help with the method of component elimination, or with analysis of component combination. The joint can be examined and evaluated from different perspectives and can cause a problem from an efficiency aspect. The assembly operations often need to add additional materials for joints (such as screws, rivets or welding filler metal). Figure 1 shows examples of sheet metal joining methods [2].

The disadvantage of joints can occur when local weakening is created in the welded joints as a result of microstructure changes, which consequently cause impairment of mechanical properties [3]. All these possibilities can influence the safety of the joint, and they usually lead to an increase in the weight of the joints in order to fulfil the required function. In addition, maintenance and replacement of components or recycling are already considered in many cases, thus, the simplicity of assembly and joints is a prerequisite for the initial design of the joint [4]. In Figure 1, sheet metal joining methods are shown [2].

Joining requirements are important for the design of joining. Generally, there are four defined types of design issues for joining [1,4,5]:The material types and properties to be joined;The geometry of the joint;The required functions that the joint has to fulfil;The production conditions of future joining.

Environmental requirements are also necessary for production purposes:Design for disassembly;Material and energy saving;Reuse of materials;Recycling of materials;Fuel economy regulation.

Therefore, there are necessary requirements to investigate the new alternative and progressive methods of material assembly in the automotive industry. The shaping and friction drilling techniques are now among the most promising future alternatives for the creation of joining, with the possibilities of rapid and economic solutions for producing joints with less joining parts [6,7,8].

From this point of view, friction or thermal drilling becomes an innovative possibility when the materials that should to be joined have very different chemical and mechanical properties. In recent years, Fe metal and light metals created the basis for metals in the automotive industry. The new special techniques are extremely important for joining in automotive applications and play a key role in the research area. The effects of joint modification stresses and melting and thermal processes can create brittle inter-metallic compounds during joining phases that may result in brittle joints [7,8,9,10].

Friction drilling (FD) is an efficient, quick and clean technology process that can make reliable automotive components with similar resistant properties and mass reduction. The most famous firms, such as Audi©, use special technologies such as the Flow Drill Screwing (FDS^®^; from the Swedish company EJOT) process. The builders of Mazda© and Honda have also used new friction stir welding (FSW, which has similar inputs at the beginning steps of joining as friction drilling), mainly composed of aluminium and steel in the manufacture of the chosen vehicle components [11,12,13,14]. Green manufacturing has gained significant interest of the scientific community in almost all fields of mechanical and production engineering. It is known that we need to use a lubricant in drilling operations, and during drilling operation chips are generated as waste. Thermal drilling technology belongs to chipless technology and uses minimum lubricants, thus presenting one possibility of decreasing production waste and subsequently protecting the environment [11,12,15,16].

## 2. Friction Drilling Method

The friction thermal drilling process belongs to a non-conventional process, in which holes, bushings and collars can be created in the used material. [8,9,13,14]. Friction drilling is a metalworking process that occurs just above the recrystallization temperature of the testing material. The result is grain deformation during the friction processing. The grains start to recrystallize, the microstructure has equated properties, and the temperature prevents the metal from excessive work hardening [6,8,13,14,17]. Heat generation depends on more conditions, such as the surface speed of the tool and testing material’s mechanical properties, for example, plasticity. According to research [1,9,14], speed has the biggest influence and effect on the surface roughness of the tested material. The high speed of the drilling tool increases the frictional heat that causes higher plasticity, and this is why better results occur with less dimensional errors [1,11,12,13,18]. It was demonstrated that with the increase in the tool speed from 2500 to 4500 rpm, the surface roughness of the tested material decreases from 0.536 to 0.341 µm [6,7,13,14,17,18,19].

The quality of the created holes is affected by technological parameters, such as speed; friction forces; created heat during the friction drilling process; geometry of the tool; and the material properties, for example, the conductivities of the material [20,21,22,23,24].

The 3D finite element modelling of the friction drilling process was also explained by many authors [9,14,25] to show the material flow process, the temperature change during the process, and the stresses and strains of the tested materials, which are very difficult for experimental measurements. The friction drilling tool is designed in many variants, but it mainly has two different parts: a conical part, which opens the aperture and softens the tested material, and a cylindrical part that determines the final geometry of the hole or bushing. The geometry of the friction tools protects them from the initial wear, and it increases the tool life. To decrease the wear of the drilling tool, a small amount of lubricant or paste is also usually applied on tool before the friction drilling operation to protect the tool from early wear and to prevent the material transfer from the tested material to the tool [6,8,25,26]. The significant burring on exiting the hole depends on the thermal friction tool [19,25,27]. As the Flowdrill tool drills into the material, one part of the displaced material from the holes or bushings forms a collar around the upper surface of the base metal, and the second part of the displaced material forms a bushing in the lower surface of the base metal [8,23]. The bushings, created by thermal drilling operations and following tapping operations, increase the stiffness joints of the automobile parts. Figure 2 presents thermal drilling and tapping processes [23].

The thermal drilling process also has many advantages over the traditional drilling processes. The first advantage is that it is not necessary to cool the base metal. In conventional drilling processes, lubricant reduces the friction between the tool and the tested material, and heat facilitates chip evacuation. The friction drilling process is a clean process and does not create chips. Moreover, the excess material or burrs, occurring in the end of bushing, can be used in the next operation during the creation of thread in holes in complex and inaccessible tubular geometries. [8,12,18,20,23,28,29].

After thermal drilling of the holes in the materials, the form tapping process can be used for threads. Threads have widespread utilization in many mechanical joining applications, and such a combination of drilling and tapping operations can be easily automated in work lines. In the manufacturing process there are usually two methods used to create the inner threads [27,30,31,32,33,34,35]:Made by cutting, the thread is obtained as in many other machining operations by chip removal.Threads production is achieved by forming or tapping operations. The threads may be produced by the cold forming operation, which involves rolling, deformation of the raw material under cold working conditions, and tapping, in order to achieve a good joint of similar strength to conventional drilling, but simultaneously avoiding the use of nuts (and even screws) in some cases.

## 3. Experimental Method and Details

In the frame of experiments, in the Department of Process and Environmental Engineering, Faculty of Mechanical Engineering, TU of Kosice, the suitability of thermal drilling technology for the chosen Al alloy materials was verified. The most important questions and tasks of friction drilling were solved with methods to optimize and improve the geometrical shape of future bushings and collars. The experiments were designed on the sequence of steps from the simplest method to evaluation by methods of electron microscopy. The first step of evaluation was visual evaluation of the created bushings and collars and then by optical microscope and scanning electron microscope (SEM, Tescan, Brno, Czech) evaluations. Finally, energy dispersive X-ray spectroscopy (EDS, Tescan, Brno, Czech) were used to define and control material composition. The experiments were carried on a Flott P2 drilling machine (Flott, Remscheid, Germany) with a speed range of 450–4000 rpm, as shown in Figure 3. Experimental works on the machine were conducted manually. Three speed ranges were used for experiments in intervals: A1 = 1470 min^−1^, A2 = 2530–2550 min^−1^ and A3 = 3200–3430 min^−1^. The Flowdrill tool (Flowdrill, Utrecht, Netherlands) has standard geometry, divided on the conical section and the cylindrical section, the shoulder and the shank. The Flowdrill drilling tools are made of tungsten carbide in cobalt matrix and have been selected for experiments [4,24].
Drill 1: type—Flowdrill short, diameter Ø 7.3 mm (M8).Drill 2: type—Flowdrill short, with a milling cutter of Ø 7.3 mm (M8) and lubrication paste Flowdrill type FDKS.Forming tap M8; rpm during tapping: 600–680 min^−1^; lubrication oil Flowdrill type FTMZ.

Properties of the FD tool: hardness (Rockwell hardness scale—HRA): 89–93.5; maximum working temperature: 900 °C.

The base metals (square pipe and double layers of material) were placed on top of each other and were fastened in the jig, as shown in Figure 4.

The material selected for this study was Al alloy (AlMgSi) EN AW-6060 according to the European standard STN EN 755-2:2016 [36], and its chemical composition was as displayed in Table 1 and its mechanical properties as displayed in Table 2. The dimension and the type of base metals were a square pipe (prism) with a 30 × 30 × 2 mm^3^ thickness and a longitude of 300 mm.

### 3.1. Experimental Setup

The planned experiment consisted of two parts:Visual evaluation of the testing samples;Metallographic and microscopy evaluation.

The authors also investigated the deformation of the processed materials and checked the created material joints, formation and occurrence of cracks by microscope evaluation. Firstly, the bushings and collars from the tested one-layer sample were evaluated. Then, the bushings and collars from two layers materials were created, because the produced holes and bushings allowed for the creation of a greater and longer space, providing, for example, better future utilization as a material screw joint.

#### 3.1.1. Visual Evaluation

Visual evaluation of the experiment was conducted according to standard STN EN ISO 16348 [37]—“evaluation by visual control”. The testing samples were prepared by changing the technological conditions, that is, by changing the testing speed. The photos of the sorted tested samples according drilling speed are shown in Figure 5.

The shape qualities of the created collars are different and depended on the drilling speed. Regularly split, torn collars were created at lower speeds—1470 min^−1^. The higher speed (2530 to 3430 min^−1^) created better integrity of the shapes of the collars. The friction-drilling tool has a milling part, with enables the collar to be cut down and creates more possibilities for joints in the following assembly operations. An example of hole after cutting the collar is also shown in Figure 5.

Visual assessments of created bushings and collar from one material thickness are presented in Figure 6. The bushing with tapped threads and the sealed samples with and without collars are displayed in Figure 6.

Examples of the created bushings from two material layers are in shown Figure 7. The base metals were stacked on top of each other without pressing and were only touched in the jigs. The created bushings were without cracking, but with smaller holes in the bottom and had a more conical shape. After cross-cutting of the tested samples, some materials leaked in and pressed between the gaps (layers) of sheets, as can be seen in Figure 7.

The authors wanted to point out the imperfections made in the thermal drilling process. The most frequently occurring imperfections are listed in Figure 8 for visual evaluation. They can be eliminated by correctly setting the process parameters, by the right selection of material, and by automating the process, where repeatability is guaranteed.

#### 3.1.2. Metallographic and Microscopy Evaluation

Metallographic macroscopic evaluation was the next step. This consisted of evaluation of the shape of the bushing and radius of the collar and bushing inside the tapped threads that were created, as shown in Figure 9. The shape of the cross-cut bushing is without cracks, and the collar is rounded. The tapped threads are regular, and errors were not observed by light microscope (OLYMPUS BX, Tokyo, Japan) FM or with digital photo apparatus (OLYMPUS C-4040, Tokyo, Japan).

The microstructures from the thread part of the bushing are shown in Figure 10. The compressed grains are observed at the edges of the threads.

The details of the deformed grains from the left and right side of the cross-cut collar are shown in Figure 11. Considerable deformation of the grains is observed when viewed in detail. The grains are deformed in the direction of flow of the material when forming the collar. The grains are most deformed in the rounded part of the collar.

The macrostructures and microstructures of the middle part of the two layers of the bushing from both sides are also shown in Figure 11, where the mixing border of two base metals in the bushing wall are shown. Material flow between the two layers of the mixed material can be seen. The following microstructures illustrate the mixing of the material during the drilling operation and creation of the bushing. The mixed materials form a permanent joint, that is, the body of the bushing.

The details of the investigated material joint and the place of the two mixed materials are shown in Figure 12. Further observations were made via scanning electron microscope (SEM) and energy dispersive X-ray spectroscopy (EDS) analysis.

The investigated place of the middle part of the bushing is shown in Figure 13. The details of the created crack and mixing area (two layers) of the base metals in the cylinder part of the bushing are displayed in Figure 14. Figure 15 presents further details of Figure 14, where the deformation and small cracks occurred in front of the division of the material into two layers. These small cracks occurred only in the observed area of investigation. The details of the shape of the small cracks are shown in Figure 16.

The measured area of small crack creation is shown in Figure 17, where the width of the measured area was 125.6 μm and the height was 133.2 μm in front of the determined place.

The chosen areas for investigation by scanning electron microscope (SEM) are shown in Figure 18. The chemical compositions in the chosen areas are revealed by EDS spectra, wher: spectrum 19 (Figure 19) is the area of the formation of a large crack and division of two non-mixed layers, spectrum 20 (Figure 20) is the area of small cracks, and spectrum 21 (Figure 21) is the area of mixed materials, without cracks and errors, near the hole of the bushing.

#### 3.1.3. Hardness Test

The hardness Vickers test (HV) was conducted according the standard STN EN ISO 6507-1 [38] and standard STN EN ISO 6507-3 [39] in the hardness apparatus PMT3, diamond pyramide, load 200 g. The measuring places were chosen from both sides as 1—basic unheated material, 2—part of the FD bushing with threads, the left side, 3—part of the FD bushing with threads, the right side, and 4—basic unheated material. The measuring results are shown in Figure 22, where the sample with measured places and the measured Vickers HV values can be observed.

Table 3 shows the average measured Vickers hardness values of more chosen places from three test specimens (that is, basic non-heat-affected material, heat-affected material, the collar and the bushing). The measured results show that the basic, non-heat-affected material have higher values than the hardness values of the formed bushing.

## 4. Discussion

A friction thermal drilling operation is a non-conventional process of creating holes and bushings of various types of materials, and it enables the creation of joining from one or more layers of material. The aim of this research was to evaluate the quality of created collars and bushings dependent on technological parameters from the easiest way (visual evaluation) to the most specific evaluation (electron microscope evaluation). The tested samples were divided into two groups: tested samples consisting of one layer (square pipe) and tested samples with double layers (square pipe and Al sheet).

According to visual evaluation, the quality of the created bushings, holes and collars fulfilled the requirements for future joint utilization. The created thread, achieved by tapping operations, can also be utilized as a future screw joint. The bushing with threads had a longer and bigger area for screwing, and the future joint will be stronger, fixed and safety. The quality of the created bushings and collars depend on the technological parameters. The higher the speed, the better the bushings and collars. The created bushings and threads in the bushings have the best shape and strength properties at higher speeds of the drilling machine at 3400 rpm, as shown in Figure 5.

The SEM analysis performed in Figure 13, Figure 14, Figure 15, Figure 16, Figure 17 and Figure 18 shows a cross-section of the material, from mixed to divided parts. Our main attention was paid to the place where the material began to mix, that is, where it formed the first joint and cracks and showed an indication of the direction of mixing. The presence of some impurities on the base of Fe, C and O was confirmed by EDS spectra, but their occurrence is very small and does not effect the formation of the joint, as shown in Figure 19, Figure 20 and Figure 21.

The thermal drilling operation is a chipless operation with minimum of lubricant utilization. The design of the tool enabled us to quickly decrease the created temperature. This technology can be classified as environmentally friendly technology.

## 5. Conclusions

In the presented contribution, the last view of modern progressive technology was introduced, that is, thermal drilling by Flowdrill. The experiments were prepared and oriented for the evaluation of Al alloy material parameters at various defined technological conditions, focused on the quality of the created collars, bushing and threads, their visual, metallography evaluations and SEM and EDS evaluation. The obtained results from material testing with various thicknesses were satisfactory in comparison with the works of the cited authors in the article.

The bushings (created with or without the threads) in base material elongate the joining width, and the possible future joints become stronger and safer. In addition, the bushings can be utilized as assisting parts in non-demount joints.

The main advantages of the thermal drilling operation from the obtained testing results can be listed as:Reduction of the number of required technological operations and simplified process for preparing collars, bushings and holes.Use of material with various thicknesses and types for creating joints.Possibility to use mixed materials with various chemical and mechanical properties.Reduction of the amount additional materials such as nuts, welding electrodes, etc.The short production time of 2–6 s depends on the thickness and type of the used material.Increase in the usable thickness of the material by the creation of bushing by up to three times in comparison with the original thickness of the base material.The thermal drilling operation reduces or even completely removes waste material (chips) from the production. Material from the drilling holes is transported into collar and bushing.By rolled threads in the bushing or by removal of the collar from the upper joining place, it increases the variability in the use of joints.Reduction of the number of required pieces of equipment for manual and/or automated production.The possibilities that chipless technology offers, i.e., energy saving, low cost, simplification of forming and cutting technological operations and undemanding for special equipment and without negative environmental impacts, are important factors from the environmental perspective.

## Figures and Tables

**Figure 1 materials-13-05007-f001:**
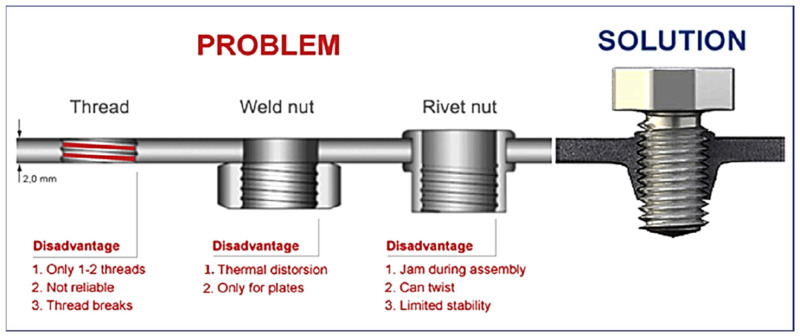
Sheet metal joining methods [2].

**Figure 2 materials-13-05007-f002:**
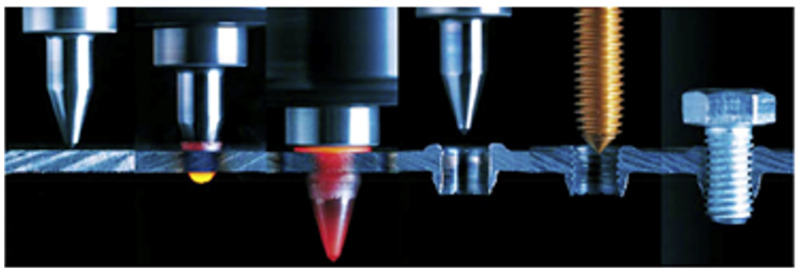
Thermal drilling and tapping processes [23].

**Figure 3 materials-13-05007-f003:**
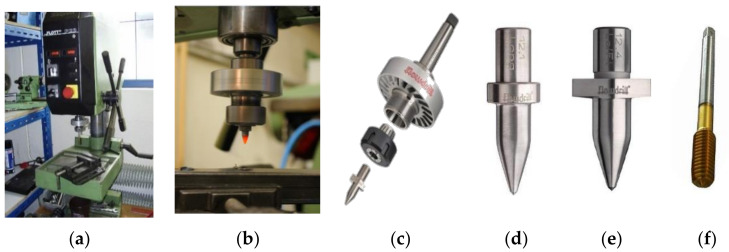
The used drilling machine and tools: (**a**) Flott P2; (**b**) details of the fastened friction drilling (FD) tool; (**c**) the FD tool with a cooling disc; (**d**) FD tool short; (**e**) FD tool short with a milling cutter; (**f**) tapping tool.

**Figure 4 materials-13-05007-f004:**
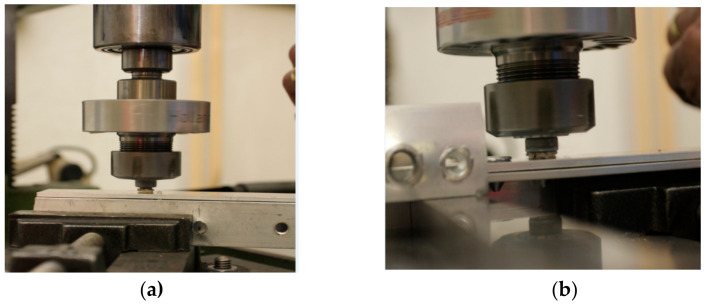
Fastening of the base metals in the jig: (**a**) square pipe, (**b**) two material sheets.

**Figure 5 materials-13-05007-f005:**
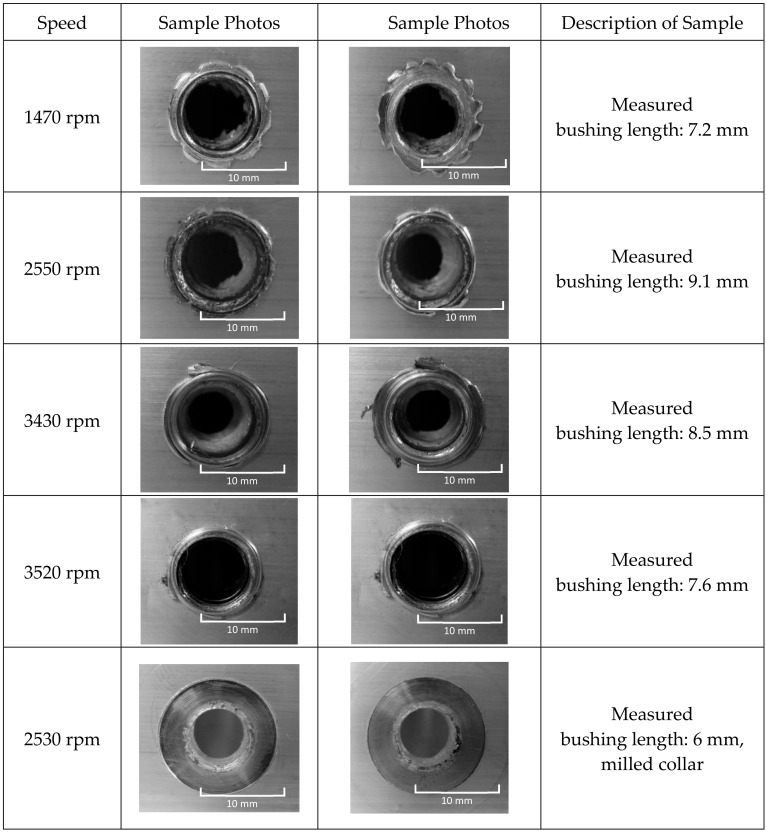
Visual evaluation of collars at different speeds of tools; view from the top.

**Figure 6 materials-13-05007-f006:**
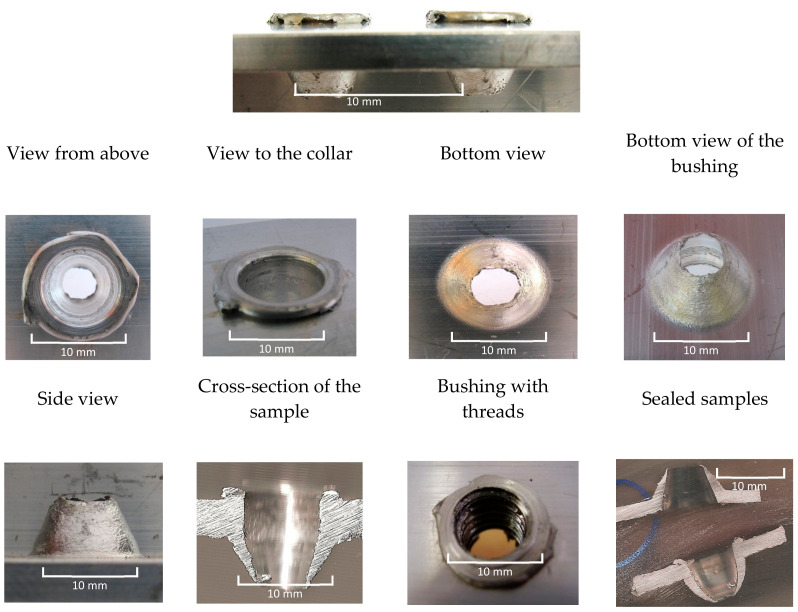
Visual evaluation of collars and bushings from one layer of the AL alloy material—Al Mg Si.

**Figure 7 materials-13-05007-f007:**
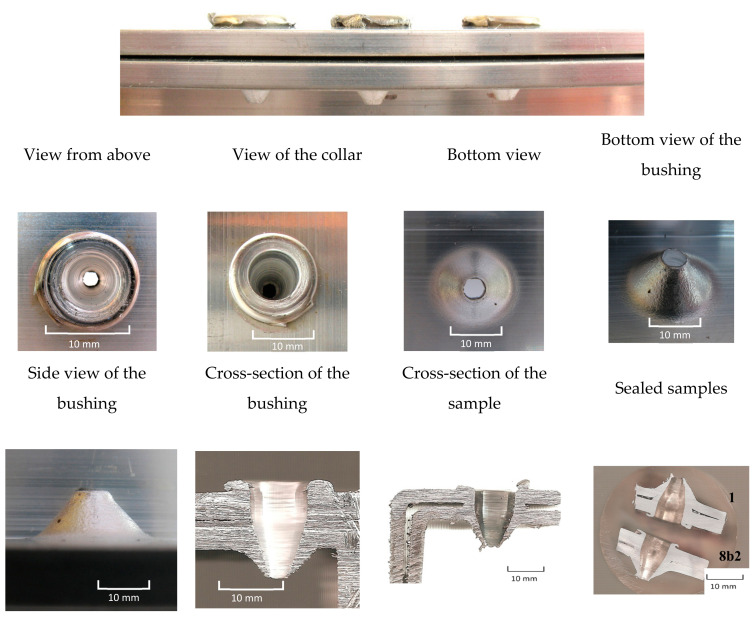
Visuals evaluation of the collar and bushing created from double layers of the Al alloy—Al Mg Si.

**Figure 8 materials-13-05007-f008:**
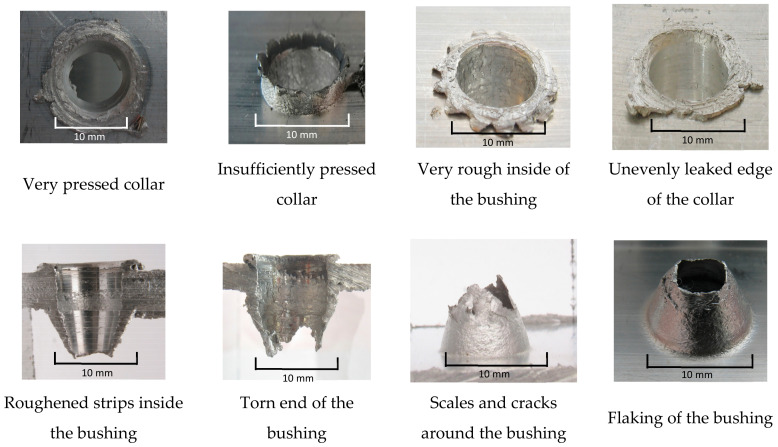
The possible imperfections during the creation of samples.

**Figure 9 materials-13-05007-f009:**
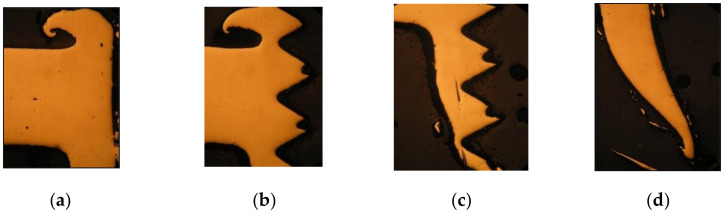
Visual evaluation of the macrostructure of the tested sample, Magn. 50×. (**a**) Collar and part of the polygon-shaped bushing. (**b**) Collar and cut threads in the polygon. (**c**) Cross-cut of the end of the bushing with threads. (**d**) Cross-cut of the end of the bushing.

**Figure 10 materials-13-05007-f010:**
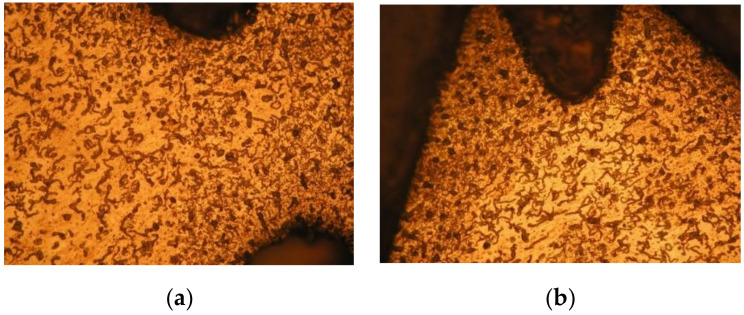
Details of the microstructures of the tapped threads, Magn. 200×. (**a**) Collar deformation and thread. (**b**) End of thread.

**Figure 11 materials-13-05007-f011:**
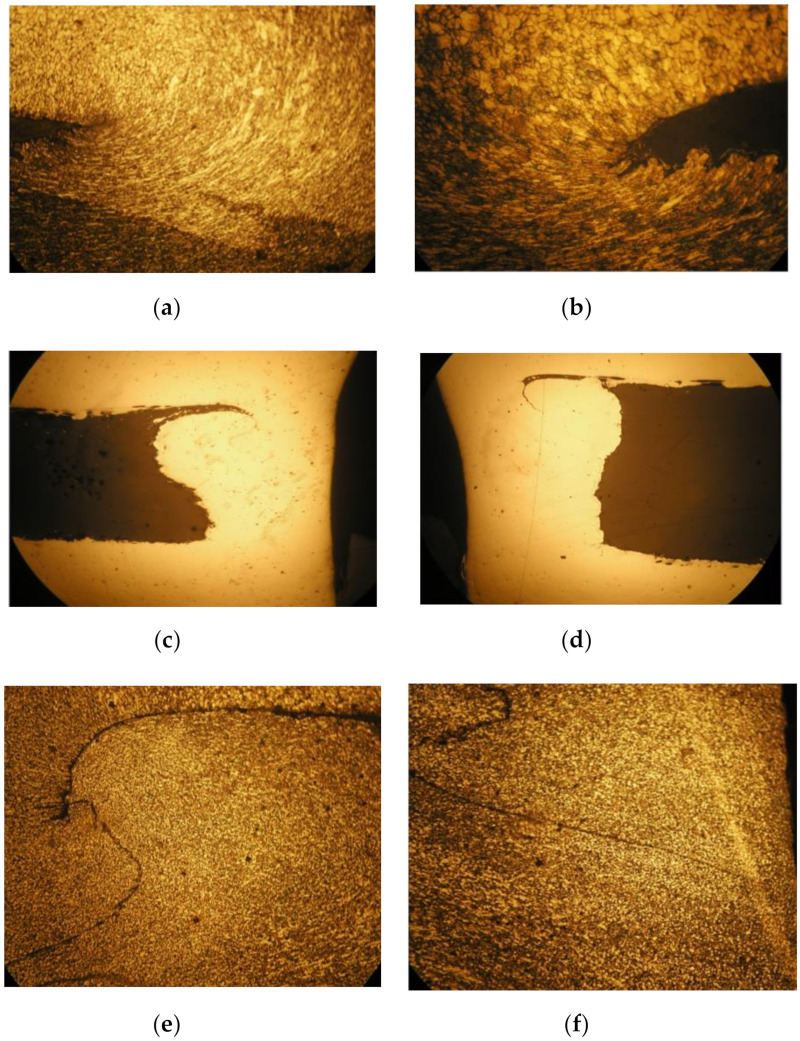
Macrostructures and microstructures of the created collar and bushing from two layers of the testing material. (**a**) Microstructure of collar – left part, Mag. 200×. (**b**) Microstructure of collar – right part, Mag. 200×. (**c**) Macrostructure of bushing left part, Mag. 50×. (**d**) Macrostructure of bushing right part, Mag. 50×. (**e**) Microstructure of mixed left bushing, Mag. 100×. (**f**) Microstructure of mixed right bushing, Mag. 100×.

**Figure 12 materials-13-05007-f012:**
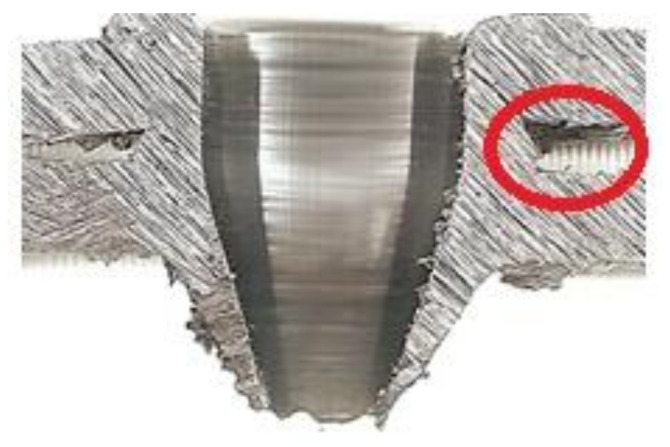
Place for SEM and EDS analysis: mixed part and initiation of the crack from inside the bushing.

**Figure 13 materials-13-05007-f013:**
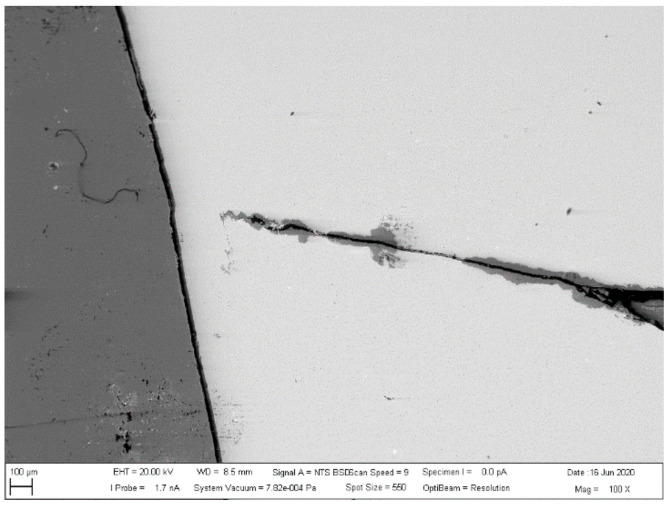
Creation of the crack in the joint.

**Figure 14 materials-13-05007-f014:**
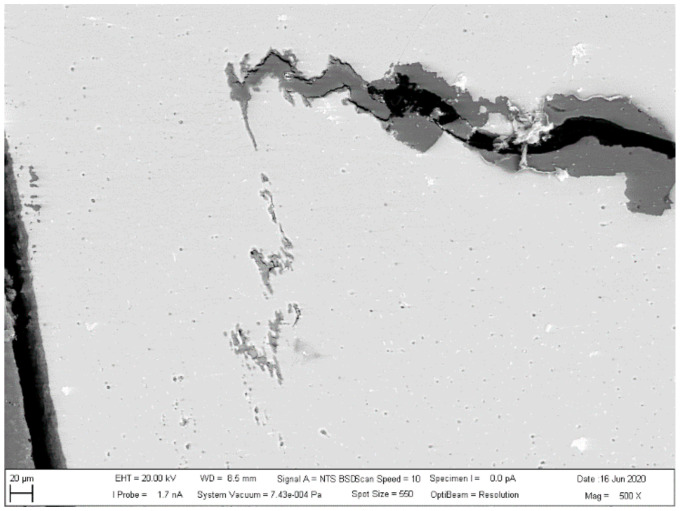
Details of cracks and mixing of materials.

**Figure 15 materials-13-05007-f015:**
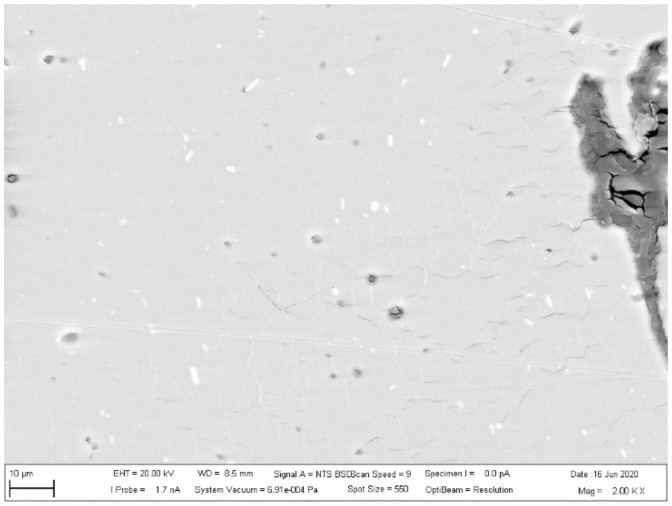
Creation of small cracks in front of the division of the material into two layers.

**Figure 16 materials-13-05007-f016:**
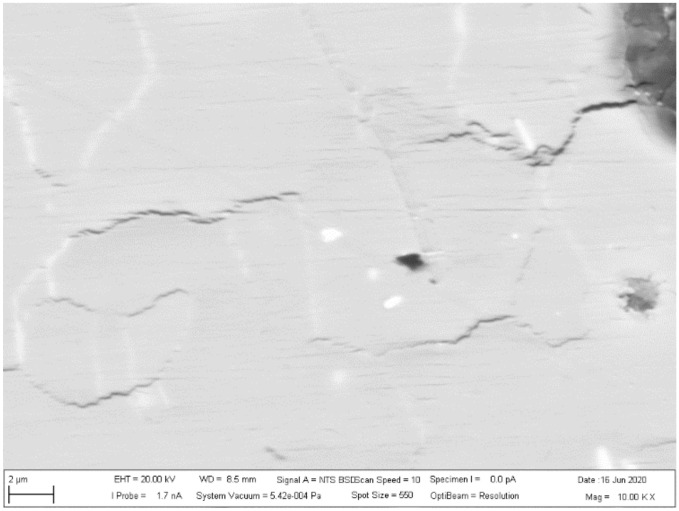
Details of small cracks.

**Figure 17 materials-13-05007-f017:**
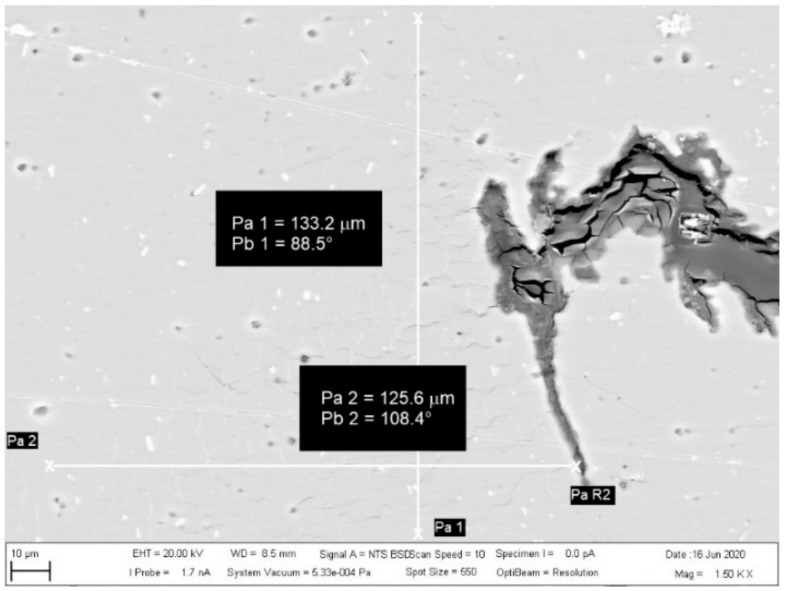
The measured area of the determined place with small cracks.

**Figure 18 materials-13-05007-f018:**
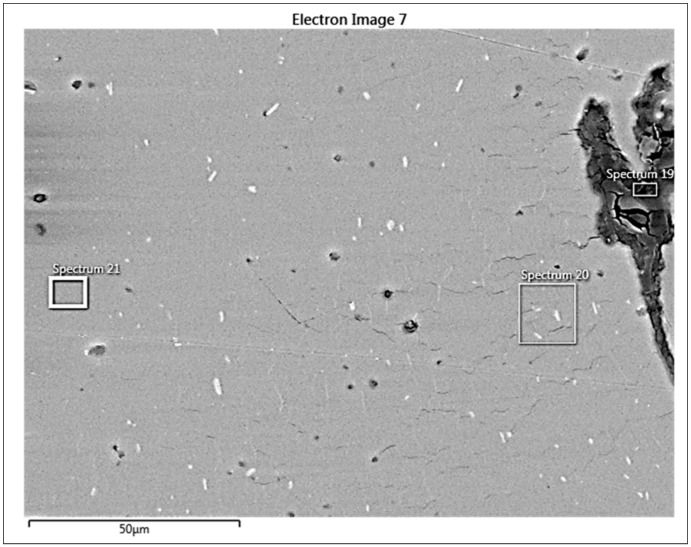
The chosen places for EDS evaluation.

**Figure 19 materials-13-05007-f019:**
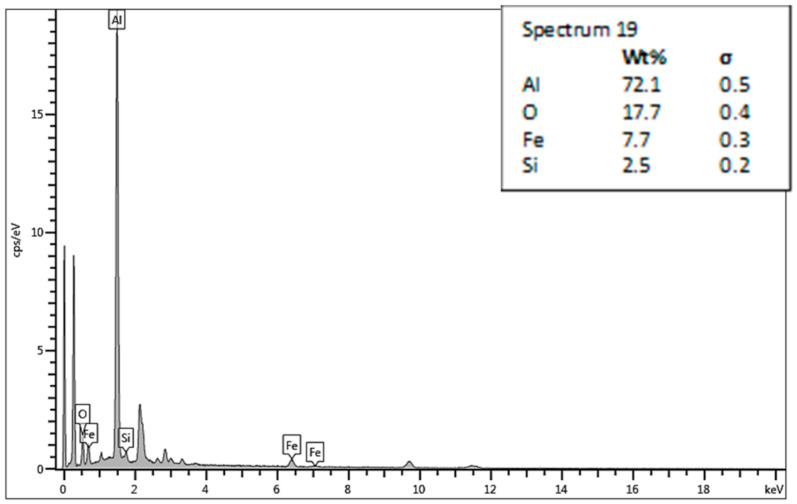
Energy dispersive X-ray (EDS)—spectrum 19.

**Figure 20 materials-13-05007-f020:**
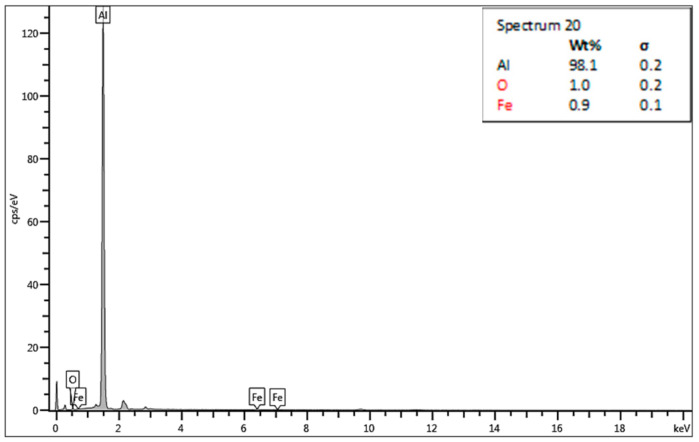
Energy dispersive X-ray (EDS)—spectrum 20.

**Figure 21 materials-13-05007-f021:**
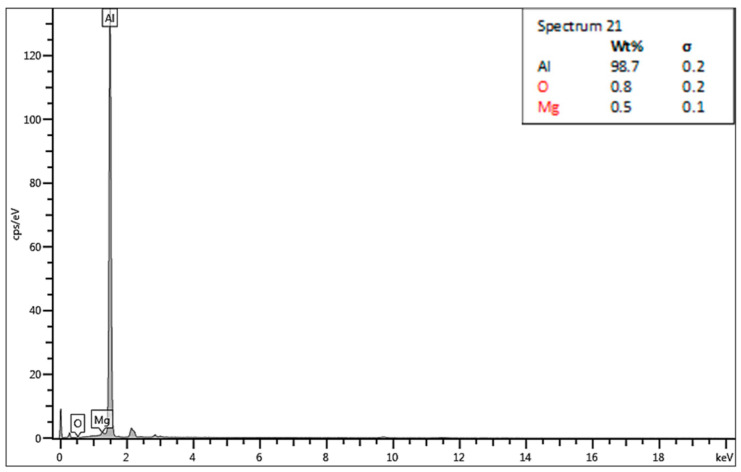
Energy dispersive X-ray (EDS)—spectrum 21.

**Figure 22 materials-13-05007-f022:**
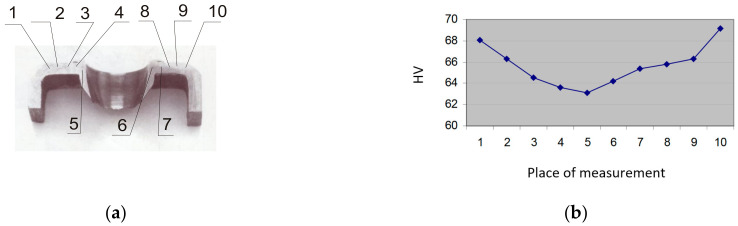
Microhardness: (**a**) Measuring places 1 and 10—basic material; 2, 3 and 8, 9—heat-affected material; 4 and 7—deformed material under the collar; and 5 and 6—deformed material in bushing. (**b**) Measured values.

**Table 1 materials-13-05007-t001:** Chemical composition of the base metal.

Chemical Element	Al	Fe	Cu	Si	Mg	Other Elements
**Content (%)**	98.8	0.1	0.1	0.3	0.4	0.3

**Table 2 materials-13-05007-t002:** Mechanical properties of the base metal.

Tested Sample	*R*m * (MPa)	*A*_80_ * (%)
AlMgSi	120–215	6–14

* *R*m—limit of strength; *A*_80_—A elongation at 80% deformation.

**Table 3 materials-13-05007-t003:** Vickers microhardness values of the Al alloy.

Measuring Place	1	2	3	4	5	6	7	8	9	10
Microhardness Sample 1	68.08	66.3	64.5	63.6	63.1	64.2	65.4	65.8	66.3	69.2
Microhardness Sample 2	73.5	-	62.8	53.4	53.4	62.8	-	-	-	76.5
Microhardness Sample 3	62.6	-	56.9	45.6	45.7	60.1	-	-	-	62.6

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
