# Peer review of "New Findings in the Field of Thermal Drilling of Aluminum Alloys"

_materials, 2020, doi:10.3390/ma13215007_

Round 1

Reviewer 1 Report

The study reports an interesting investigation on determining the effect of thermal drilling on aluminum alloys. Few items are to be addressed before publication. The manuscript can be improved by considering the following: 

  • Please update the abstract with the results reported in this paper. Some quantifiable results to address the objectives of this paper.
  • In abstract, you can avoid mentioning “This contribution was made in cooperation with the Technical University in Košice with the company U. S. Steel Košice, s.r.o.”. Its better to mention this in the acknowledgement.
  • Please check the grammatical mistakes and sentence construction throughout the manuscript.
  • Please rephrase line 31-32, “Therefore, the exist theory of…”
  • Authors may add a schematic of the friction drilling method for more clarification.
  • From the literature review, it is not clear why the present work is needed. Authors need to explain the limitations of the current techniques and the identified research gap to be mentioned.
  • Please provide the other mechanical properties of the material in Table 2, e.g. YM, UTS etc.
  • Please put scale bar in all the tables (3, 4, 5, 6,7,8,9)
  • Authors need to explain in details the results obtained from the hardness test, SEM results and correlating them with the drilling speeds.
  • As one of the purposes to join the materials together, authors are requested to test and report the joint strength.

Author Response

Response to Reviewer 1 Comments

Dear reviewer,

Thank you very much for your reviewing of our article and your kind help to improve our article.

According to your requirements, we tried to fulfil the requirements and questions:

Point 1. 

- Please update the abstract with the results reported in this paper. Some quantifiable results to address the objectives of this paper.

Response 1:

We have corrected the content of the abstract and added the missing requirements. (in red)

Point 2. 

An abstract, you can avoid mentioning “This contribution was made in cooperation with the Technical University in Košice with the company U. S. Steel Košice, s.r.o.”. Its better to mention this in the acknowledgement.

Response 2:

We have pasted the company's thanks at the end of the acknowledgement. (in red)

Point 3. 

Please check the grammatical mistakes and sentence construction throughout the manuscript.

Response 3:

We tried to check our mistakes and also we asked the editor for the translation help.

Point 4.

Please rephrase line 31-32, “Therefore, the exist theory of…”

Response 4:

We have corrected the whole paragraph of the article. (in red)

Point 5.

Authors may add a schematic of the friction drilling method for more clarification.

Response 5:

We added two schematic of the friction drilling method, Fig. 1 and Fig. 2. (in red)

Point 6.

From the literature review, it is not clear why the present work is needed. Authors need to explain the limitations of the current techniques and the identified research gap to be mentioned.

Response 6:

We corrected the method and explained that we wanted to evaluate a quality of created collars, bushings after change of technological parameters and after changes of material layers ( one layer and two layer materials) (in red)

Point 7.

Please provide the other mechanical properties of the material in Table 2, e.g. YM, UTS etc.

Response 7:

We corrected and explained the shortcuts of the mechanical properties of material in Table 1 (in red)

Point 8.

Please put scale bar in all the tables (3, 4, 5, 6,7,8,9)

Response 8:

We put scale bar in all the tables and according the requiremen of seconf reviewer , we changed the tables to figures (in red)

Point 9.

Authors need to explain in details the results obtained from the hardness test, SEM results and correlating them with the drilling speeds.

Response 9:

We explained the detailes  of hardness Vicker test., SEM results and dependence of quality according to speed. (in red)

Point 10.

As one of the purposes to join the materials together, authors are requested to test and report the joint strength

Response 10:

Thank you for your comment, but in this article, the aim was not to compare the tensile tests of the material screw joint, but to compare the quality of the perforated hole, bushing and collar under different technological conditions. The comparison of tensile tests and their evaluation will be the basis for another contribution.

Reviewer 2 Report

This manuscript shows experimental research on new findings in the field of thermal drilling of aluminum alloys. However, major correction is required before make a decision or publish in Material.

1- There are grammatical errors throughout the entire manuscript; the authors need to revisit the manuscript very carefully to refine the language. For example page 3/17; First advantage is in no need for cooling of testing material!

2- Page 1/17, local weakening can occur due to the microstructure changes which results in the impairment of mechanical properties. Therefor, you should change the statement as following using the suggested reference;

“The disadvantages of the joints can occurs, when local weakening is created in the welded joints as a results of microstructure changes which consequently cause the impairment of mechanical properties [ref]….

[Ref] EFFECT OF FILLER METALS ON THE MECHANICAL PROPERTIES OF DISSIMILAR WELDING OF STAINLESS STEEL 316L AND CARBON STEEL A516 GR 70

3- “Testing material” should be changed to “ base metal” throughout the entire manuscript.

4- The base metal is AlMgSi then why there is no Mg in the composition data in Table 1?

5- What are Rm and A80 in Table 2? Please identify them.

6- Table 3 - 9; first must be changed to Figure 3 - 9, then show the scale in all of them to better understanding of the size of experiment results.

7- In Figures 10-11, EDS elemental results (wt%) are not clear to read. Please make them much clearer using black color or larger font size.

8- In Figure 13, points 2 and 3 are almost same locations. Then why their microhardness values are different? please give the reasonable explanation.

9- In Table 10; please identify the samples 1-3 and locations 1-10. It would be much better if you show the locations in a figure or schematic illustration.

10- In the conclusion section, no need to show any reference. Also conclusion must be specific and extracted from the major finding of research. This conclusion is very long and you would better to re-write it by showing the main conclusion in a clear way.

Author Response

Response to Reviewer 2 Comments

Dear reviewer,

Thank you very much for your reviewing of our article and your kind help to improve our article.

According to your requirements, we tried to fulfil the requirements and questions:

Point 1.

There are grammatical errors throughout the entire manuscript; the authors need to revisit the manuscript very carefully to refine the language. For example page 3/17; First advantage is in no need for cooling of testing material!

Response 1:

We tried to check our mistakes and also we asked the editor for the translation help.

Point 2.

Page 1/17, local weakening can occur due to the microstructure changes which results in the impairment of mechanical properties. Therefor, you should change the statement as following using the suggested reference;

“The disadvantages of the joints can occurs, when local weakening is created in the welded joints as a results of microstructure changes which consequently cause the impairment of mechanical properties [ref]….

[Ref] EFFECT OF FILLER METALS ON THE MECHANICAL PROPERTIES OF DISSIMILAR WELDING OF STAINLESS STEEL 316L AND CARBON STEEL A516 GR 70

Response 2:

We corrected the text and sentence and added the reference into list of references. ( in red)

Point 3.

“Testing material” should be changed to “ base metal” throughout the entire manuscript.

Response 3:

We corrected and change in the whole article the “Testing material” into “base metal”  (in red).

Point 4.

The base metal is AlMgSi then why there is no Mg in the composition data in Table 1?.

Response 4:

We corrected and put the values of the missing composition Mg. We made mistake during the preparation of the Table. ( in red)

Point 5.

What are Rm and A80 in Table 2? Please identify them

Response 5:

We corrected and explained the meaning of Rm and A80. (in red)

Point 6.

6- Table 3 - 9; first must be changed to Figure 3 - 9, then show the scale in all of them to better understanding of the size of experiment results

Response 6:

We corrected and changed the Tables 3-9 to Figures. (in red)

Point 7.

In Figures 10-11, EDS elemental results (wt%) are not clear to read. Please make them much clearer using black color or larger font size.

Response 7:

We corrected and made larger the elemental results (wt%). (in red)

Point 8.

- In Figure 13, points 2 and 3 are almost same locations. Then why their micro-hardness values are different? please give the reasonable explanation.

Response 8:

We corrected the figure of measured places in mentioned Figure 13, now as Figure 22 (in red)

Point 9.

 In Table 10; please identify the samples 1-3 and locations 1-10. It would be much better if you show the locations in a figure or schematic illustration.

Response 9:

We corrected  and changed the Figure 22 which is more corresponded with “Table  10“, now as Table 3 and explained the measured places.(in red)

Point 10. 

In the conclusion section, no need to show any reference. Also conclusion must be specific and extracted from the major finding of research. This conclusion is very long and you would better to re-write it by showing the main conclusion in a clear way.

Response 10:

We corrected, changed and shorted the conclusion. (in red)

Reviewer 3 Report

This study introduces joining and pushing technique by thermal drilling method. However, to accept this paper for publication the following comments need to be addressed

  • The introduction section needs to be improved by citing new and related articles of the current journal.
  • The proficiency of the language needs a more improvement in the manuscript
  • What is the meaning of Rm and A80[%] in table 2. It is better to write the abbreviation of these terms firstly.
  • In Table 10, which places you measured?
  • The discussion section needs to be modified. There is a leakage of discussion.
  • The Conclusion should be more concise.

The manuscript can be accepted after minor revision

Author Response

Response to Reviewer 3 Comments

Dear reviewer,

Thank you very much for your reviewing of our article and your kind help to improve our article.

According to your requirements, we tried to fulfil the requirements and questions:

Point 1. 

The introduction section needs to be improved by citing new and related articles of the current journal.

Response 1:

We have corrected the introduction section and added the citing (in red).

Point 2. 

The proficiency of the language needs a more improvement in the manuscript

Response 2:

We tried to check corrected our mistakes in manuscript and also we asked the editor for the translation help. (in red)

Point 3. 

What is the meaning of Rm and A80[%] in table 2. It is better to write the abbreviation of these terms firstly.

Response 3:

We corrected and explained the shortcuts of the mechanical properties of material in Table 1 (in red).

Point 4. 

In Table 10, which places you measured?

Response 4:

We explained the details of hardness Vickers test and the measured places (in red).

Point 5. 

The discussion section needs to be modified. There is a leakage of discussion.

Response 5:

We modified the discution section (in red).

Point 6. 

The conclusion should be more concise.

Response 6:

We modified and shortage the conclusion (in red).

Round 2

Reviewer 1 Report

Thanks for the updated version of the manuscript. Most of the comments are addressed. 

Reviewer 2 Report

Revision was done accordingly and manuscript in this format is acceptable to publish in Materials.